# Reducing Sialylation Enhances Electrotaxis of Corneal Epithelial Cells

**DOI:** 10.3390/ijms241814327

**Published:** 2023-09-20

**Authors:** Bryan Le, Kan Zhu, Chelsea Brown, Brian Reid, Amin Cressman, Min Zhao, Fernando A. Fierro

**Affiliations:** 1Department of Ophthalmology, University of California, Davis, CA 95616, USA; bynle@ucdavis.edu (B.L.); minzhao@ucdavis.edu (M.Z.); 2Department of Cell Biology and Human Anatomy, University of California, Davis, CA 95817, USA

**Keywords:** electrotaxis, corneal wound healing, sialic acid, sialylations, corneal epithelial cells

## Abstract

Corneal wound healing is a complex biological process that integrates a host of different signals to coordinate cell behavior. Upon wounding, there is the generation of an endogenous wound electric field that serves as a powerful cue to guide cell migration. Concurrently, the corneal epithelium reduces sialylated glycoforms, suggesting that sialylation plays an important role during electrotaxis. Here, we show that pretreating human telomerase-immortalized corneal epithelial (hTCEpi) cells with a sialyltransferase inhibitor, P-3FAX-Neu5Ac (3F-Neu5Ac), improves electrotaxis by enhancing directionality, but not speed. This was recapitulated using Kifunensine, which inhibits cleavage of mannoses and therefore precludes sialylation on N-glycans. We also identified that 3F-Neu5Ac enhanced the responsiveness of the hTCEpi cell population to the electric field and that pretreated hTCEpi cells showed increased directionality even at low voltages. Furthermore, when we increased sialylation using N-azidoacetylmannosamine-tetraacylated (Ac4ManNAz), hTCEpi cells showed a decrease in both speed and directionality. Importantly, pretreating enucleated eyes with 3F-Neu5Ac significantly improved re-epithelialization in an ex vivo model of a corneal injury. Finally, we show that in hTCEpi cells, sialylation is increased by growth factor deprivation and reduced by PDGF-BB. Taken together, our results suggest that during corneal wound healing, reduced sialylated glycoforms enhance electrotaxis and re-epithelialization, potentially opening new avenues to promote corneal wound healing.

## 1. Introduction

Corneal wound healing is a complex, dynamic process in which a host of different signals are integrated to efficiently heal the wound. One such signal is the endogenous wound electric field, which serves as an important cue to guide cell migration [1,2]. Notably, the rate of corneal wound healing is strongly correlated with the strength of the wound electric field [3,4].

Glycosylations of macromolecules including proteins, lipids, and even small RNAs are a key modification modulating cell function [5,6]. Thus, the diversity of glycoforms is critically important to tune the response of cells to environmental cues, including differentiation, secretion, survival, and migration [7,8,9]. At the terminus of many glycans are negatively charged monosaccharides known as sialic acid (N-acetylneuraminic acid; Neu5Ac), which affects a host of glycan–protein interactions [10]. In unwounded corneas, the epithelium of the central cornea is highly sialylated [11]. Upon wounding, there is a downregulation of sialylated glycans [12,13]. This is partly caused by downregulating transcription of key enzymes that add sialic acid to glycans in the Golgi apparatus, known as sialyltransferases [12,13,14]. 

It remains unknown how sialylations impact electrotaxis during corneal wound healing. The concurrent downregulation of sialylations with the generation of the endogenous electric field suggests that downregulating sialylations would enhance electrotaxis. However, it was previously shown that the enzymatic removal of sialylations using neuraminidases (also known as sialidases) impaired electrotaxis of macrophages [15]. Interestingly, while most neuraminidase-treated macrophages exhibited no preference for either the cathode or anode, a small population of cells reversed their directionality to the cathode, suggesting that modulating sialylated glycans may tune the response of cells to the electric field [15]. 

To determine the role of sialylation during electrotaxis, we modulated the sialylations of human telomerase-immortalized corneal epithelial (hTCEpi) cells through metabolic glycoengineering and measured migration during electrotaxis. To modulate sialylations, we pretreated hTCEpi cells with P-3Fax-Neu5Ac (3F-Neu5Ac), which is a potent inhibitor of sialyltransferases [8,16] or with Kifunensine, a compound that inhibits mannosidases [17,18,19]. Due to the potential implications of promoting corneal wound healing, the major goal of these studies was to determine the role of sialylations in electrotaxis of corneal epithelial cells.

## 2. Results

### 2.1. Reducing Sialylation Enhances Cathodal Migration of hTCEpi Cells

To test whether 3F-Neu5Ac would effectively reduce sialylation, we measured *Sambucus nigra* (SNA) lectin binding, which recognizes sialylations in an α-2,6 configuration and to a smaller extent in an α-2,3 configuration, through flow cytometry. As expected, 3F-Neu5Ac significantly reduced sialylation in hTCEpi cells (Figure 1A,B). We found that treatment with 3F-Neu5Ac (100 μM) caused a significant reduction in proliferation after 4 days (Figure 1C). Thus, all experiments were performed subsequently using 3F-Neu5Ac at 100 µM for no more than 2 days. 

To determine how sialylation impacts electrotaxis, we pretreated hTCEpi cells with 3F-Neu5Ac for 48 h before applying a DC electric field at physiologically relevant strength (100 mV/mm). We first determined that there were no significant differences in directionality or speed in the absence of an electric field (Appendix A). When applying an electric field, hTCEpi cells pretreated with 3F-Neu5Ac showed enhanced cathodal migration during electrotaxis (Figure 1D and Appendix A). We calculated both directionality and speed every 30 min and detected a significant (*p* < 0.05) increase in directionality within the first hour of applying an electric field (Figure 1E). However, this difference gradually narrowed over time, likely due to an increasing number of control cells responding to the electric field and migrating towards the cathode. Importantly, we did not detect any significant differences in cell speed, at any time during the recording (Figure 1F).

### 2.2. Reducing Sialylation May Enhance the Sensitivity of hTCEpi Cells to the Electric Field

Since hTCEpi cells pretreated with 3F-Neu5Ac showed a significant increase in directionality within the first hour of electrotaxis, we postulated that hTCEpi cells may have become more sensitive to the electric field. To test this, we measured the directionality at 15, 30, and 60 min after applying a weak electric field at 30 mV/mm. Control hTCEpi cells showed a significant increase in directionality after applying an electric field for 30 and 60 min, but not after 15 min (Figure 2A). In contrast, hTCEpi cells pretreated with 3F-Neu5Ac showed a significant difference in directionality as early as 15 min (Figure 2B). Furthermore, hTCEpi cells showed relatively low directionality but remained biased towards the cathode during electrotaxis (Figure 3C and Appendix A). We detected a significant increase in directionality by 60 min after applying the electric field (Figure 3D). We did not detect significant differences in speed at any time point.

To further determine if hTCEpi cells pretreated with 3F-Neu5Ac are more sensitive to the electric field, we reversed the polarity of the applied electric field (100 mV/mm) every 30 min during electrotaxis. We measured the change in directionality after 10 min at every frame. We reasoned that pretreated hTCEpi cells will show stronger shifts in directionality when the polarity reverses. Indeed, we observed that pretreatment with 3F-Neu5Ac increased the number of hTCEpi cells changing directionality within 30 min, as compared to control cells (Figure 2F). Furthermore, hTCEpi cells pretreated with 3F-Neu5Ac had fewer cells that were non-responsive to the polarity reversal. Taken together, our results suggest that reducing sialylation enhances electrotaxis in part by sensitizing the cells to the electric field.

### 2.3. Increasing Sialylation Impairs Electrotaxis

Since we determined that reducing sialylation enhances electrotaxis, we next asked whether increasing sialylation would impair electrotaxis. We first determined that treating hTCEpi cells with 100 μM of Ac4ManNAz (a precursor of sialic acid [20]) for 48 h showed a small but significant increase in SNA lectin staining (Figure 3A,B). Strikingly, we observed that hTCEpi cells pretreated with Ac4ManNAz showed a rounded morphology that appeared to lack polarity normally observed in control cells (Figure 3C). When an electric field was applied, Ac4ManNAz-pretreated hTCEpi cells polarize within 30 min. Interestingly, some Ac4ManNAz-pretreated cells polarized towards the anode, which was maintained up to 4 h after applying the electric field. We detected a significant decrease in directionality and speed during electrotaxis, when compared to control cells (Figure 3D–F and Appendix A). Taken together, our results suggest that increasing sialylations affects both directional sensing and motility during electrotaxis.

### 2.4. Pretreating hTCEpi Cells with Kifunensine Enhances Electrotaxis

Although 3F-Neu5Ac reduces sialylation, it may inadvertently increase the expression of other polysaccharide structures. To further confirm that reducing sialylation enhances the directionality of hTCEpi cells during electrotaxis, we pretreated hTCEpi cells with Kifunensine to promote high-mannose N-glycans that preclude sialylation of N-glycans [17,18]. We first determined that treating hTCEpi with Kifunensine at 20 μg/mL for 48 h significantly enhanced ConA lectin binding, which is specific for oligomannose-type N-glycans, and significantly reduced SNA lectin binding (Figure 4A–D). Consistent with our results with 3F-Neu5Ac, we found that, during electrotaxis, Kifunensine also significantly increased cell directionality, but not speed, as compared to control (Figure 4E,F). However, unlike pretreatment with 3F-Neu5Ac, we did not detect a significant difference in directionality within the early time points after applying an electric field (Figure 4F and Appendix A). Only after 2.5 h of electrical stimulation, cells pretreated with Kifunensine showed a significant increase. These results confirm that reducing sialylations promotes the directionality of hTCEpi cells under an electric field and suggest that the exact dynamics are dependent on the types of induced N-glycoforms. 

### 2.5. Pretreating Enucleated Eyes with 3F-Neu5Ac Promotes Re-Epithelialization in an Ex Vivo Model of Corneal Injury

Since reducing sialylations enhanced electrotaxis, we next asked if pretreating murine eyes with 3F-Neu5Ac could enhance corneal re-epithelialization ex vivo. To test this, we enucleated murine eyes and placed them in standard culture media supplemented with 3F-Neu5Ac for 24 h before wounding (Figure 5A). The eyes were allowed to heal for an additional 24 h before fluorescein staining. We found a significant increase in wound closure in eyes pretreated with 3F-Neu5Ac (Figure 5B,C). Our findings suggest that modulating sialylation may be a novel approach towards promoting corneal wound healing. 

### 2.6. hTCEpi Cells Modulate Sialylations in Response to PDGF-BB and Growth Factor Starvation

Finally, we sought to determine if hTCEpi cells regulate sialylations in response to environmental cues. We have previously identified that serum starvation and platelet-derived growth factor BB (PDGF-BB) increased or decreased sialylations, respectively, in human bone-marrow-derived mesenchymal stem/stromal cells (MSCs) [8]. To determine if a similar response is present in hTCEpi cells, we starved hTCEpi cells and measured SNA lectin binding, through flow cytometry. We observed a small but significant increase in SNA lectin staining (Figure 6A,B). This may in part be attributed to a significant increase in the gene expression of ST6GAL1, which codes for the sialyltransferase that adds sialylations in an α-2,6 conformation (Figure 6C). We did not detect significant differences in the expression of ST3GAL1 or ST3GAL4, which codes for sialyltransferases that add sialylation in an α-2,3 conformation. Finally, we detected a significant decrease in sialylations after hTCEpi cells were treated with the growth factor PDGF-BB for 24 h (Figure 6D,E). However, we did not observe significant differences in the expression of sialyltransferases (Figure 6F). It is therefore possible, that other sialyltransferases or neuraminidases are regulated to modulate sialylated glycans in response to PDGF-BB.

## 3. Discussion

Our results demonstrate that sialylation plays an important role during electrotaxis. When we reduced sialyation using 3F-Neu5Ac or Kifunensine, we detected a significant increase in directionality during migration in an electric field. We infer that reducing sialylation may increase the sensitivity of hTCEpi cells to the electric field. Conversely, when we increased sialylation, hTCEpi cells showed impaired directionality and speed during electrotaxis. In support, pretreating murine eyes with 3F-Neu5Ac promoted re-epithelialization in an ex vivo wound healing model.

Upon injury, the cornea epithelium generates an endogenous wound electric field and concurrently downregulates sialylations along the wound edge [12,13,14]. Given that biological processes are often coordinated, our findings suggest that reducing sialylation is critical in promoting corneal wound healing. Although we showed that 3F-Neu5Ac enhanced re-epithelialization ex vivo, the molecular mechanism remains unclear. It is important to note that the wound electric field peaks quickly after wounding before decreasing and plateauing over time [4]. This reduction may allow for desialylated corneal epithelial cells to respond to the electric field due to its increased sensitivity. In addition, enhancement in corneal wound healing may be attributed to galectin-3 binding, which has been shown to enhance wound healing [21], sialylation-mediated immunoregulation [22], or perhaps to regulating the activity of ion channels and pumps [23,24]. It remains important to evaluate the therapeutic potential of 3F-Neu5Ac in vivo.

Although we show that reducing sialylation enhanced electrotaxis in hTCEpi cells, it was previously shown that reducing sialylation using neuraminidases from *Clostridium perfingens* impaired electrotaxis in other cell types, such as in fibroblasts and Chinese hamster ovary (CHO) cells [25]. The differences may in part be attributed to the different approaches used to reduce sialylations. 3F-Neu5Ac is a fluorinated analog of Neu5ac that strongly binds to sialyltransferases to inhibit its catalytic activity. Thus, the inhibitory effect is exerted during the biosynthesis of sialylated glycans. Conversely, neuraminidase treatment acts extracellularly to remove sialylations. Neuraminidase activity can be limited through modification of the glycan hydroxyl group, unrecognized linkages, and/or steric hindrance [26]. Thus, it remains highly likely that different biomolecules are desialylated, resulting in differential impacts on electrotaxis. It will be interesting to determine whether neuraminidases and 3F-Neu5Ac differentially impact electrotaxis in hTCEpi cells. 

It was previously identified that macrophages, which normally migrate to the anode, reversed their directionality after being infected by Salmonella [15,27]. This shift in directionality was at least partially attributed to neuraminidases secreted from the bacteria. Given the reported decrease in α-2,3 sialylations, it corroborates our finding that reducing sialylations enhances cathodal migration during electrotaxis. This further supports the notion that cells actively regulate sialylations to tune their biological response to the extracellular environment. 

Mechanistically, the impact of sialylation on the directionality during electrotaxis may be attributed to integrin function, which plays a critical role in regulating the directionality of cells during electrotaxis [28]. It is of note that Integrin α5, which is expressed on hTCEpi cells [29], promotes cathodal migration of cells. Given that de-sialylation of N-glycans on integrins α5 and β1 enhances integrin activity [21,30,31], it is plausible that integrin activity suggests that reducing sialylations may promote cathodal migration by enhancing integrin α5 activity. When sialylations are increased, it may reduce the signals to migrate towards the cathode, resulting in the observed impaired electrotaxis.

## 4. Materials and Methods

### 4.1. Animals

All animal use was reviewed and approved by the Institutional Animal Care and Use Committee. C57BL/6J mice were purchased from Jackson Laboratories. All mice were maintained under standard animal housing conditions. 

### 4.2. Cell Culture

Human telomerase-immortalized corneal epithelial (hTCEpi) cells were cultured using EpiLife^®^ medium supplemented with 1% EpiLife-defined growth supplement (EDGS) and 1% penicillin and streptomycin. Cells were maintained at 37 °C with 5% CO_2_. Only cells between passages 60 and 80 were used. To reduce sialylation, hTCEpi cells were treated with 100 μM of 2,4,7,8,9-pentaacetyl-3Fax-Neu5Ac-O2Me (3F-Neu5Ac) or with 20 μg/mL of Kifunensine for 48 h. To increase sialylation, the culture media was supplemented with 100 μM of N-Azidoacetylmannosamine-tetraacylated (Ac_4_ManNAz) for 48 h. Prior to experimentation, cells were washed with PBS, and the culture media was changed to standard culture media. 

### 4.3. Flow Cytometry

Prior to analysis, hTCEpi cells were seeded at 5 × 10^4^ cells per well in a 6-well plate before treatment with 3F-Neu5Ac (100 μM), Ac_4_ManNAz (100 μM), Kifunensine (20 μg/mL), PDGF-BB (10 ng/mL), or starved (0.1% EDGS) for 2 days. hTCEpi cells were then trypsinized and centrifuged at 200× *g* for 5 min. To measure sialylated glycoforms, we used FITC-conjugated Sambucus nigra (SNA) lectin (Vector Labs, Burlingame, CA, USA) diluted 1:50 in PBS. To measure oligomannose-type N-glycans, we used FITC-labeled concanavalin A (ConA) also diluted 1:50 in PBS. ConA binds preferentially to high-mannose and hybrid-type N-glycans (REF). hTCEpi cells were incubated with the indicated lectins at 4 °C for 30 min. hTCEpi cells were washed with PBS before being measured using the Attune NxT flow cytometer. 

### 4.4. Electrotaxis

Electrotaxis experiments were performed as previously described [32]. Briefly, hTCEpi cells were seeded in electrotactic chambers with FNC Coating Mix^®^ (Athena Enzyme Systems, cat#0407). The culture media was supplemented with 100 μM of 3F-Neu5Ac, 20 μg/mL of Kifunensine, or Ac4ManNAz for 48 h. The culture media was changed to standard culturing conditions prior to experimentation. A direct current (DC) electric field was applied through agar salt bridges with Ag/AgCl electrodes in Steinberg’s solution at indicated voltages for 3 to 4 h. Images were captured every 5 min using a Carl Zeiss Observer Z1 inverted microscope and the MetaMorph NX program. Speed and directionality were calculated using the plugin MTrackJ on ImageJ. Trajectory plots were generated using the Chemotaxis and Migration Tool (ibidi). 

### 4.5. Ex Vivo Corneal Wound Healing

Eight-week-old C57BL/6J mice were euthanized using CO_2_ and cervical dislocation. Then eyes were carefully removed using fine spring scissors (Fine Science Tools, Foster City, CA, USA) and placed in DMEM/F12 culture media supplemented with 1% penicillin/streptomycin. For each experiment, one eye served as control and the other eye served as treatment. The enucleated eyes were pretreated with DMSO (vehicle, Control) or with 3F-Neu5Ac for 24 h prior to wounding. We generated circular corneal wounds using a 1 mm trephine, an Algerbrush II (Accutome Inc., Malvern, PA, USA), and an ophthalmologic scalpel (Medical Sterile Products, Rincon, Puerto Rico). The wounded area was marked using a Ful-Glo fluorescein sodium ophthalmic strip (Akron Inc., Washington, DC, USA). Images were taken immediately after wounding and after 24 h using a Zeiss Lumar V12 microscope with an Axiocam MRm camera and an EXFO X-cite 120 fluorescent illumination system. We determined the wound area using ImageJ and calculated the percentage of wound healing using the following equation: (original wound area − new wound area)/(original wound area) × 100

### 4.6. Gene Expression Analysis

To determine the gene expression of sialyltransferases, we extracted RNA from hTCEpi cells using the Direct-zol RNA Miniprep kit (Zymo Research, Irvine, CA, USA) following the manufacturer’s instructions. The mRNA was reverse transcribed using the TaqMan Reverse Transcription Reagents (ThermoFisher, cat# N8080234) and measured using the TaqMan Universal PCR Master Mix (Invitrogen, cat# 4304437) and TaqMan Gene Expression Assays (Invitrogen). Primers and probes can be identified by the following Assay ID: GAPDH: Hs03929097_g1; ST3GAL1: Hs00161688_m1; ST6GAL1: Hs00949382_m1; ST3GAL4: Hs00920870_m1. 

### 4.7. Statistical Analysis

Results are represented as mean with standard deviation as error bars. The number of replicates (independent experiments) is shown in each figure. Statistical analysis was performed using GraphPad Prism 10.0.2. A paired student’s t-test was used to determine statistical differences for mean fluorescent intensity, cell speed, cell directionality, and percentage of wound healing. A *p*-value of < 0.05 was considered statistically significant.

## 5. Conclusions

Here we showed that modifying sialylated glycans strongly impacts the electrotaxis of corneal epithelial cells. Reducing sialylations promotes electrotaxis, while increasing sialylations reduces it. These findings suggest that reducing sialylations may promote corneal repair, which we show using a murine ex vivo model. Finally, we show that key signals regulate sialylations of corneal epithelial cells, at least in part by transcriptional control of sialyltransferases.

## Figures and Tables

**Figure 1 ijms-24-14327-f001:**
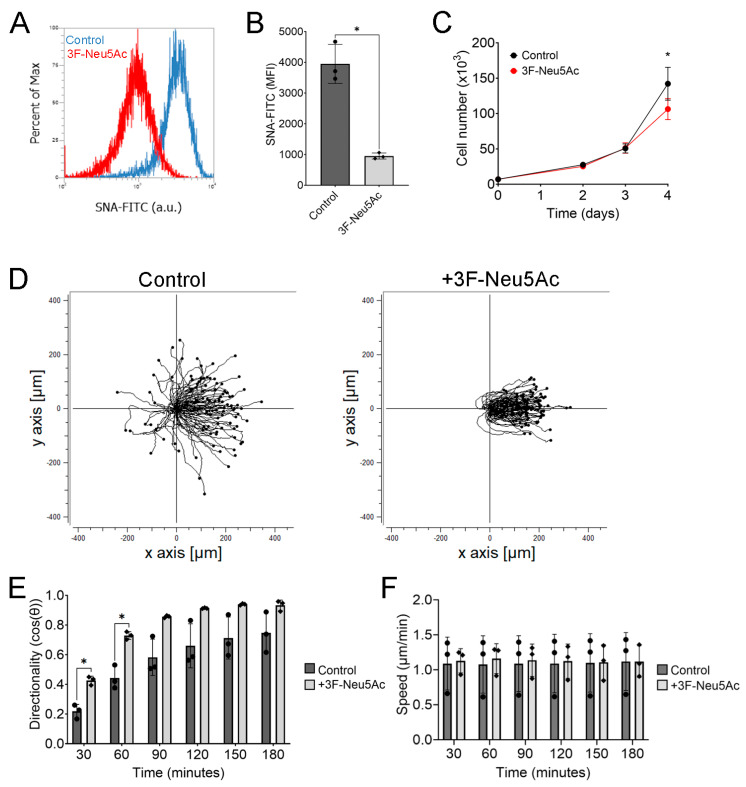
Pretreatment with 3F-Neu5Ac enhances electrotaxis of hTCEpi cells. hTCEpi cells were seeded into electrotactic chambers and pretreated with 100 μM of 3F-Neu5Ac for 48 h before electrical stimulation. (**A**) Representative histogram of hTCEpi cells treated with 3F-Neu5Ac. (**B**) Quantification of the MFI (*n* = 3). (**C**) Proliferation of hTCEpi cells cultured with or without 100 μM 3F-Neu5Ac (*n* = 5). (**D**) Representative trajectory plots of hTCEpi cells pretreated with 3F-Neu5Ac. (**E**) Quantification of directionality (*n* = 3). (**F**) Quantification of speed (*n* = 3). Bar graphs show averages with standard deviation as error bars. MFI, mean fluorescence intensity; * *p* < 0.05 as calculated using a paired Student’s *t*-test.

**Figure 2 ijms-24-14327-f002:**
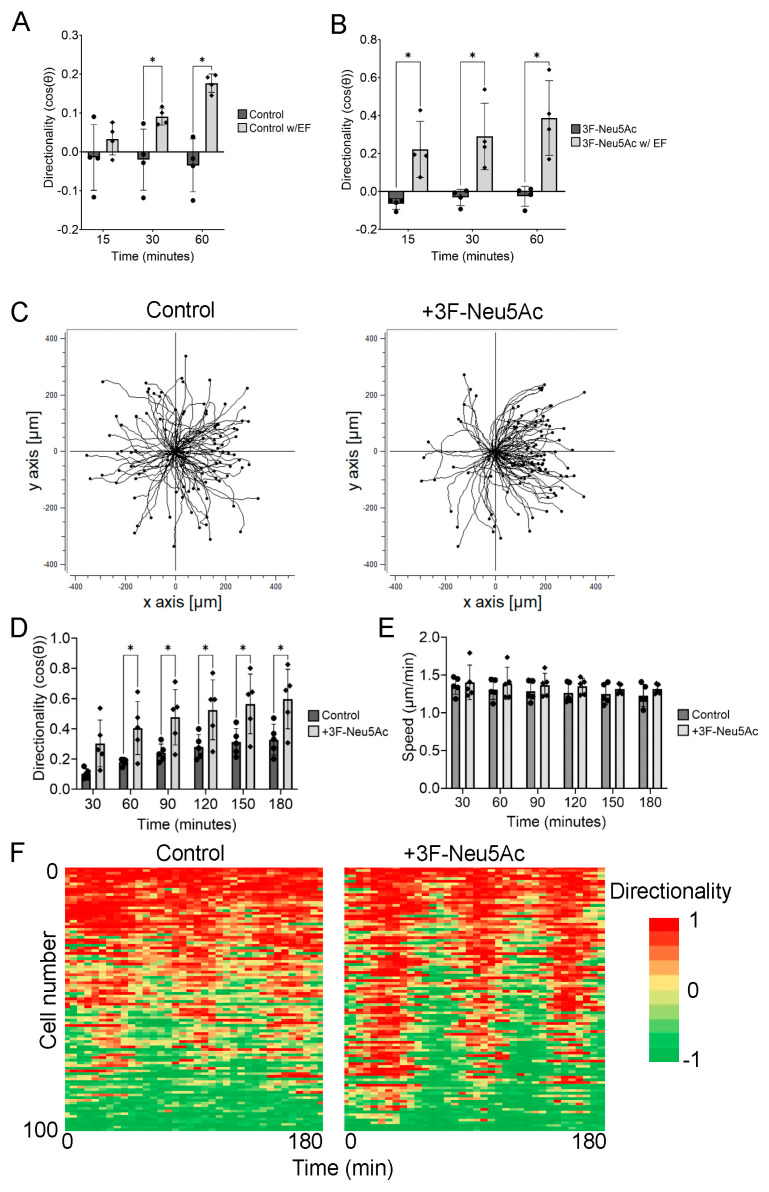
Pretreatment with 3F-Neu5Ac sensitizes hTCEpi cells to the electric field. hTCEpi cells were pretreated with 3F-Neu5Ac before a weak electric field at 30 mV/mm was applied. (**A**) Comparison of the directionality and speed of control cells at 15, 30, and 60 min with or without electric field stimulation (*n* = 5). (**B**) Directionality and speed of cells pretreated with 3F-Neu5Ac at 15, 30, and 60 min with or without electric field stimulation (*n* = 5). (**C**) Representative trajectory plots under weak electric fields. (**D**) Quantification of directionality over 3 h with electrical field stimulation (*n* = 5). (**E**) Quantification of speed over 3 h with electrical field stimulation. (**F**) Representative heatmap of directionality over time along the x-axis and cell number along the y-axis. Bar graphs show averages with standard deviation as error bars. * *p* < 0.05 as calculated using a paired Student’s *t*-test.

**Figure 3 ijms-24-14327-f003:**
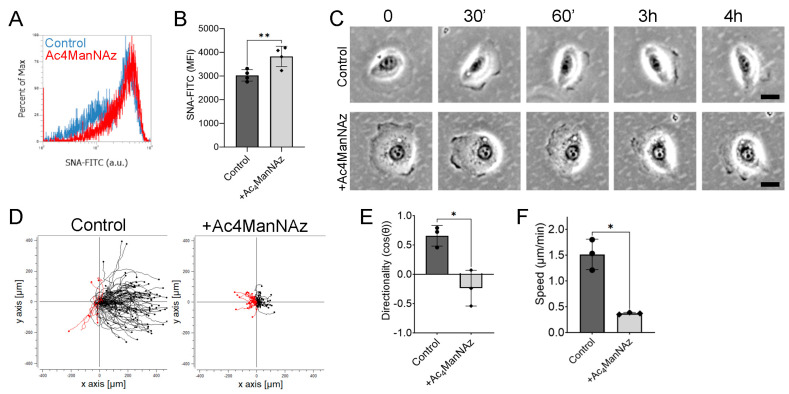
Increasing sialylation reduces electrotaxis of hTCEpi cells. hTCEpi cells were seeded in electrotactic chambers before being pretreated with 100 μM of Ac4ManNAz. (**A**) Representative histogram of cells treated with Ac4ManNAz. (**B**) Quantification of the MFI (*n* = 4). (**C**) Representative hTCEpi cell migrating in an electric field at 0, 30, 90, 180, and 240 min. (**D**) Representative trajectory plot of hTCEpi cells treated with or without Ac4ManNAz. Red trajectories indicate anodal migration. (**E**) Quantification of directionality (*n* = 3). (**F**) Quantification of cell speed after 4 h (*n* = 3). Bar graphs show averages with standard deviation as error bars. MFI, mean fluorescence intensity; * *p* < 0.05 and ** *p* < 0.01 as calculated using a paired Student’s *t*-test.

**Figure 4 ijms-24-14327-f004:**
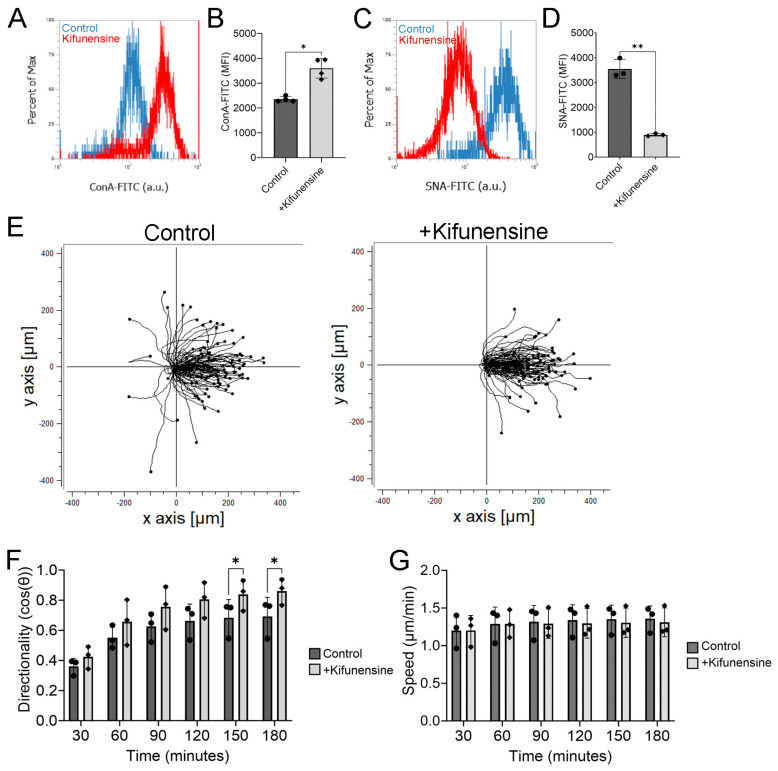
Kifunensine pretreatment enhances electrotaxis of hTCEpi cells. hTCEpi cells were pretreated with Kifunensine (20 μg/mL) for 48 h before applying an electric field at 100 mV/mm. (**A**) Representative histogram of hTCEpi cells stained with concanavalin A (ConA) lectin. (**B**) Quantification of the MFI (*n* = 4). (**C**) Representative histogram of hTCEpi cells stained with SNA lectin. (**D**) Quantification of the MFI (*n* = 3). (**E**) Representative trajectory plots of hTCEpi cells treated with or without Kifunensine in an electric field at 100 mV/mm. (**F**) Quantification of directionality every 30 min during electrical stimulation (*n* = 3). (**G**) Quantification of cell speed (*n* = 3). Bar graphs show averages with standard deviation as error bars. MFI, mean fluorescence intensity. * *p* < 0.05 and ** *p* < 0.01 as calculated using a paired Student’s *t*-test.

**Figure 5 ijms-24-14327-f005:**
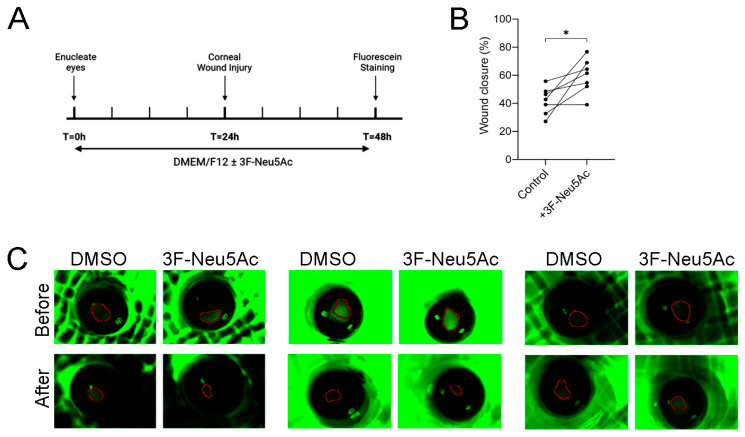
Pretreating enucleated murine eyes with 3F-Neu5Ac promoted re-epithelialization. Pairs of enucleated murine eyes were pretreated with or without 3F-Neu5Ac for 24 h before mechanical injury. (**A**) Experimental design and timeline. (**B**) Quantification of wound closure with each pair of dots representing one biological replicate (*n* = 7). (**C**) Three representative independent experiments of wounded murine eyes immediately after wounding and 24 h post-wounding. * *p* < 0.05 as calculated using a paired Student’s *t*-test.

**Figure 6 ijms-24-14327-f006:**
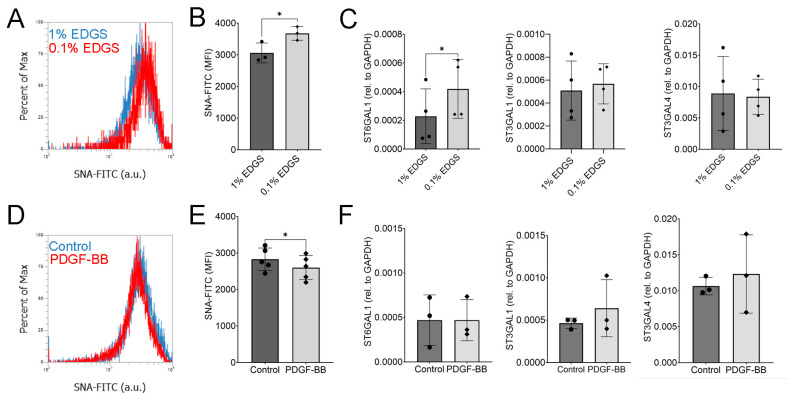
hTCEpi cells modulate sialylation in response to PDGF-BB and starvation conditions. hTCEpi cells were cultured under low serum conditions (0.1% EDGS) or with PDGF-BB (10 ng/mL) before SNA lectin staining and gene expression analysis. (**A**,**D**) Representative histograms under starvation and PDGF-BB conditions, respectively. (**B**) Quantification of MFI (*n* = 3). (**C**) RT-PCR results measuring mRNA expression of sialyltransferases (*n* = 4). (**E**) Quantification of MFI of cells stimulated with or without PDGF-BB (*n* = 5). (**F**) Expression of sialyltransferases after being treated with PDGF-BB for 1 day. Bar graphs show averages with standard deviation as error bars. MFI, mean fluorescence intensity; * *p* < 0.05 as calculated using a paired Student’s *t*-test.

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
