# Peer review of "Reducing Sialylation Enhances Electrotaxis of Corneal Epithelial Cells"

_ijms, 2023, doi:10.3390/ijms241814327_

Round 1
Reviewer 1 Report
The manuscript by Le et al described modifying sialylated glycans strongly impacts the electrotaxis of corneal epithelial cells. The manuscript has been written well and the conclusions supported by data presented.
I recommend publication in current form.
Minor changes required.
Author Response
Thank you for reviewing our manuscript.
Reviewer 2 Report
As attached

Author Response
We are very thankful for the thorough review and have introduced all recommended changes.
Reviewer 3 Report
Very good macuscript! The authors conducted rigorous research that has a potential to contribute to the literature. Article with proper scientific structure, coherent and well written. Clear aim of study, conclusions supported by the results. Although I recommend this article to the publication, some minor improvements could be made:
ln 52 - please add relevant citations
Figure D, Page 3; Figure C, Page 4 and Figure E on Page 6 - Very important figures for results understanding. It would be perfect, if authors could provide larger images. This could ensure better findings understanding.
Ln 270-274 - Authors included conclusions in the discussion method. For true scientific purity, it would be beneficial if this fragment was removed and incoporated in the "Conclusions" section.
Author Response
Thank you very much for the careful review. We introduced all suggested changes.